# Automatic Water Control System and Environment Sensors in a Greenhouse

Yousif Yakoub Hilal [1],*, Montaser Khairie Khessro [1], Jos van Dam [2] and Karrar Mahdi [2]

1   Department of Agricultural Machines and Equipment, College of Agriculture and Forestry, University of Mosul, Mosul 41002, Iraq; montaser.hussain@uomosul.edu.iq
2   Soil Physics and Land Management Group, Wageningen University & Research, 6708 PB Wageningen, The Netherlands; jos.vandam@wur.nl (J.v.D.); karrar.mahdi@wur.nl (K.M.)
*   Correspondence: yousif.yakoub@uomosul.edu.iq; Tel.: +964-77-3101-6244

**Abstract:** Iraqi greenhouses require an active microcontroller system to ensure a suitable microclimate for crop production. At the same time, reliable and timely Water Consumption Rate (WCR) forecasts provide an essential means to reduce the amount of water loss and maintain the environmental conditions inside the greenhouses. The Arduino micro-controller system is tested to determine its effectiveness in controlling the WCR, Temperature (T), Relative Humidity (RH), and Irrigation Time (IT) levels and improving plant growth rates. The Arduino micro-controller system measurements are compared with the traditional methods to determine the quality of the work of the new control system. The development of mathematical models relies on T, RH, and IT indicators. Based on the results, the new system proves to reliably identify the amount of WCR, IT, T, and RH necessary for plant growth. A t-test for the values from the Arduino microcontroller system and traditional devices for both conditions show no significant difference. This means that there is solid evidence that the WCR, IT, T, and RH levels for these two groups are no different. In addition, the linear, two-factor interaction (2FI), and quadratic models display acceptable performance very well since multiple coefficients of determination ($R^2$) reached 0.962, 0.969, and 0.977% with IT, T, and RH as the predictor variables. This implies that 96.9% of the variability in the WCR is explained by the model. Therefore, it is possible to predict weekly WCR 14 weeks in advance with reasonable accuracy.

**Keywords:** Arduino microcontroller; environment; sensors; models; greenhouses

## 1. Introduction

Agriculture is a pillar of the economic lifeline, but traditional, broad forms of agriculture are no longer able to meet the development requirements of modem agriculture; therefore, developing precision agriculture has become an inevitable trend. Water represents a natural extension of the agricultural concept. Water is vital for farm production, and more water can increase crop production [1]. Irrigation allows farmers to apply nutrients more precisely and uniformly to the wetted root volume, where the active roots are concentrated. When environments of plants are consistently maintained and kept within their comfort zone, plants are more photosynthetically efficient and can grow stress-free [2].

In arid and semi-arid zones, the main constraints limiting crop production in open fields are the scarcity and disparity in rainfall, high temperatures, extreme solar radiation, and the spread of weeds and diseases [3,4]. Since the beginning of this century, agriculture in Iraq has undergone many changes [5,6]. Agriculture was making valuable contributions to the Iraqi economy until production costs rose and farmers lacked any real support from the government. Neighbouring countries began to produce more at lower prices, and Iraqi farmers struggled to compete. Luckily, in the past few years, agriculture has become the one sector that has contributed the most to national food security, economic growth, and employment [7–9]. Greenhouses can provide high-quality products year-round with efficient production resources, including fertilizer, water, pesticides, and labour. Countries like

the United States, Europe, China, and countless others have used greenhouses to increase crop production [10–12]. The practice of producing crops in protected environments has developed rapidly, and by 2019, there were nearly 5,630,000 ha covered by greenhouses, high tunnels, low tunnels, or direct covers worldwide [13]. In Iraq, the cultivated area under greenhouses was 41,776 ha in 2019 [14].

A controlled greenhouse environment can provide suitable conditions for the optimal growth of vegetables and flowers in Iraq. Furthermore, greenhouses can be used to minimize the annual importation of vegetables. Greenhouses protect against high winds, unstable air temperature, insects, and airborne diseases. In addition to crop protection, the humidity of the air in these closed environments is considerably increased. Water productivity is increased and freshwater resources are used more efficiently [15,16].

Automated greenhouse monitoring has been addressed on a large scale from multiple perspectives and has been mostly focused on specific applications such as precision farming, irrigation, environmental control, yield prediction, and weed detection, to name a few [17]. Reference [18] explained that the interest in automated greenhouse monitoring has now reached an impressive level and noted that the interest in cultivation applications is growing exponentially. This increasing interest is also reflected in the significant advances in relevant technology such as various sensors as well as the development of cloud computing and machine learning techniques [19]. New technological improvements should allow farmers to realize their long-term expectations when using sensors in greenhouses [20]. For example, the water monitoring system improved water sustainability and reduced the daily water use of a beverage factory by 11% [21].

Unfortunately, due to poor management, loss of climate control, and overuse of water, there can be heavy losses of up to 40% for crops grown in greenhouses [13,14]. One of the major environmental stressors affecting plant growth and productivity is temperature. High-temperature stress frequently causes physiological disturbance and reduced yield and negatively affects the primary functions of the root systems [22,23].

Farmers took the agricultural traditions normally practiced in open fields in Iraq, such as using large quantities of water, and transferred them to greenhouses. Because of these traditions, water control systems and environmental sensors are not widely used, and farmers do not think that they are useful.

Studies carried out in countries neighbouring Iraq have presented many findings related to the use of automated monitoring in greenhouses under environmental conditions that differ from those in Iraq. The aims of these studies were to provide specific and non-exhaustive information on what automated monitoring should provide for greenhouse applications in their respective countries. Following the same vein, the present study aims to complement previous research by closely examining automated monitoring for greenhouse applications. The main objective of this research is to improve the methods used for water control systems and environmental sensors. Sensors monitor information related to the water levels and the general environments in greenhouses. Additionally, this paper focuses on how automated greenhouse monitoring could help meet the specific requirements of different stakeholders for several major greenhouse applications. Providing an overview of the emerging opportunities that could enhance the role of automated monitoring in Iraq by providing operational and efficient greenhouse application services will hopefully help reduce crop imports. Another goal of this study is to develop a simplified mathematical model of the water system, specifically concerning water consumption in greenhouses.

## 2. Materials and Methods

### 2.1. Test Site and Climate

The experiments were conducted in greenhouses affiliated with the College of Agriculture and Forestry, Mosul University, in the Mosul Governorate (36°23′24.1″ N 43°07′55.1″ E longitude, at an altitude of 234 m), Northern Iraq, as depicted in Figure 1. The soil was silt loam. It consisted of 17.4% Sand, 56.7% Silt, and 25.9% Clay. Soil EC and PH were 1.607 dS/m and 7.7%, respectively.

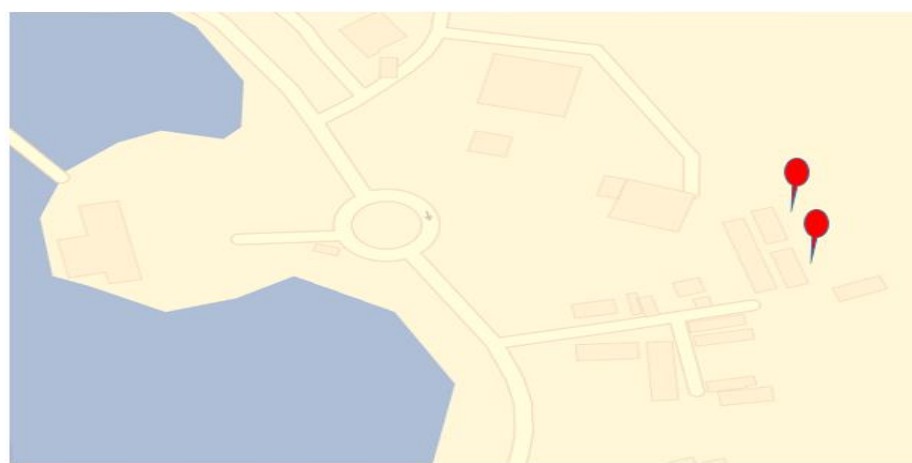

**Figure 1.** Greenhouses affiliated with the College of Agriculture and Forestry, Mosul University. https://goo.gl/maps/SuGqvhba4uQ3Tw6s5, accessed on 6 September 2021.

The study area was similar to a Mediterranean climate (a hot semi-arid climate) with a sweltering, prolonged, dry summer, brief and mild autumn, spring, and moderately wet, relatively cool winter. The ranges of rainfall, temperature, relative humidity, average cloud, and wind speed were 50–150 mm, 14–45 °C, 14–53%, 1–39%, and 14–22 km/h, respectively. Table 1 presents the average seasonal rainfall during the experimental period along with some climate characteristics.

**Table 1.** Average monthly climate at the test site.

| Months | Climate | | | |
|---|---|---|---|---|
| | Temperatures (°C) | Precipitation (mm) | Average Cloud (%) | Average Humidity (%) |
| January | 14 | 156.7 | 32 | 49 |
| February | 16 | 71.2 | 29 | 47 |
| March | 18 | 179.6 | 36 | 49 |
| April | 24 | 62.1 | 27 | 40 |
| May | 35 | 10.3 | 14 | 20 |
| June | 42 | 0 | 2 | 14 |
| July | 45 | 0 | 1 | 14 |
| August | 45 | 0 | 0 | 15 |
| September | 38 | 2.3 | 2 | 14 |
| October | 32 | 64.4 | 28 | 24 |
| November | 21 | 91.9 | 34 | 47 |
| December | 17 | 139.7 | 39 | 53 |

### 2.2. Specifications of the Plants

Cucumber F1 Bahar was one of the more suitable plants chosen largely due to high returns and a short growth period. The desired temperature for plant growth is between 15 °C and 32 °C with a growth period between 50 and 70 days. Humidity can range from 50 to 90% according to the growth stage [13].

### 2.3. Description of the Greenhouse and Control System

The greenhouse was made of galvanized steel tubes and covered with a polyethylene material. It was also equipped with a fan and a ventilation cooling system that was

used in the study. The dimensions of the greenhouse were 44 m in length, 9 m in width, and 4 m in height. The greenhouse was constructed at an elevation of 234 m. The fans were distributed inside the greenhouse according to the method described by [24] and the standard guideline of the American Society of Agricultural Engineering [25]. The polyethylene used as covering material (200 µm thick) had a low transmission coefficient compared to glass. The light transmittance was 88%, with 80% U.V. and 77% infrared.

The Arduino microcontroller system (Figure 2) was developed to control all electrical components involved in measuring temperature, humidity, and irrigation processes. The designed control board consisted of main components such as wireless communication, multiple voltage regulator circuitry (5 V, 6 V, and 9 V), and control modules integrated onto a single Printed Circuit Board. The 9 Amp/hour rechargeable battery was used as a backup when the main power supply was cut off. The Arduino was used primarily in the microcontroller system to create an interactive electronic project that could include different environmental sensors to measure temperature, humidity, and irrigation.

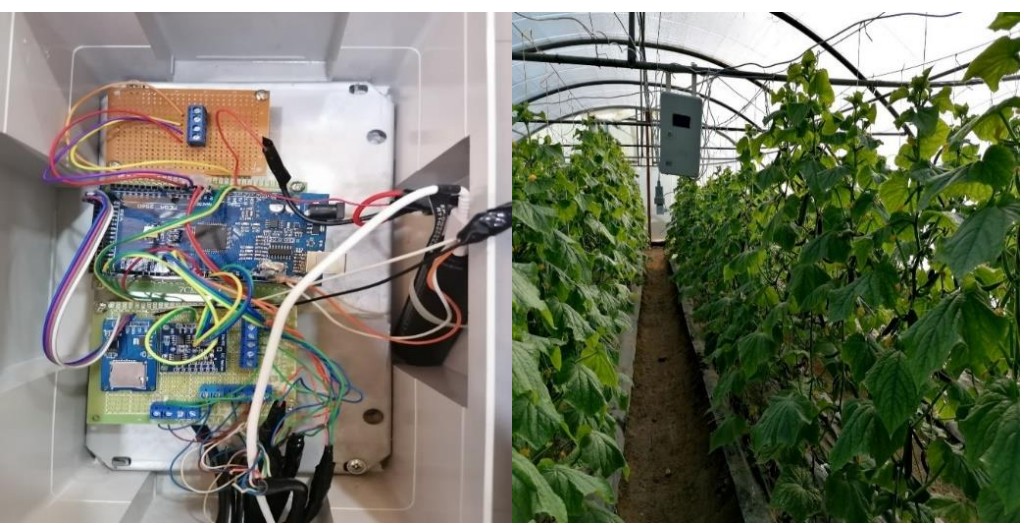

**Figure 2.** Actual prototype of the Arduino microcontroller system installed inside the greenhouse.

The Arduino microcontroller system observed daily WCR, IT, T, and RH values obtained using WCR, IT, T, and RH sensors and readings on an LCD panel. The values could be downloaded onto a USB flash drive. Four inputs and six outputs were used to control the module temperature, humidity, and irrigation processes, as described in Figure 3.

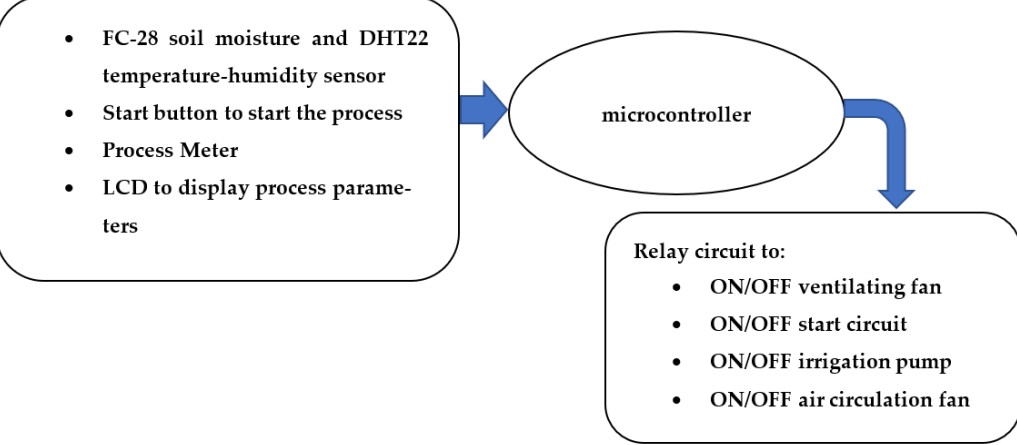

**Figure 3.** Input and output pin connections for the Arduino microcontroller system.

### 2.4. Execution of the Study and Development of Mathematical Models

The drip irrigation system used inside the greenhouse was one of the best irrigation systems suitable for greenhouse agriculture. Four irrigation pipes extended along the greenhouse. The irrigation pipe had drippers with a discharge of 3.6 L per hour, and the distance between the drippers was 20 cm. Each pipe was 40 m long and contained 200 drippers. The irrigation system was operated for two days before planting to carry out the calibration process and ensure that the drippers worked.

The traditional method used to grow this type of cucumber was followed to verify the accuracy of measuring and controlling the temperature, humidity, and irrigation processes (water consumption and irrigation time) of a new control unit (an Arduino microcontroller system). The system monitored the temperature, humidity, and irrigation processes using DHT22 and FC-28 soil moisture sensors. The system had two functions, the first one was to turn on and off the ventilating fan, air circulation fan, and the pump when the temperature, humidity, and soil moisture fell below a certain reference value. The secondary function of the system was to display the status of the temperature, humidity, and soil using an LCD. The FC-28 soil moisture sensor read soil moisture, and the DHT22 sensor read the value for temperature and humidity throughout the greenhouse as input from the system that the Arduino microcontroller system could process. Temperature, humidity, and soil moisture content were measured in three locations (at the entrance, in the middle, and at the end of the greenhouse).

The HTC-2 Digital Thermometer Hygrometer Electronic was used to measure humidity and temperature using three devices placed throughout the greenhouse. An Extech -Mo750 device was used to measure the moisture content of the soil. The details of the devices are shown in Table 2. For the purpose of verifying the accuracy of the control, the measurements obtained from the conventional devices (HTC-2 Digital Thermometer Hygrometer Electronic and Extech -Mo750) were compared with the measurements obtained from the developed control unit. Fertilization, pest control, seed quantity, greenhouse preparation, and irrigation systems were carried out as were typical for the study area [13].

**Table 2.** Specifications and range of the HTC-2 Digital Thermometer Hygrometer Electronic and Extech -Mo750.

| Extech -Mo750 | HTC-2 Digital Thermometer Hygrometer Electronic |
| --- | --- |
| Sensor Type Integrated Contact Probe | Material: ABS Size: $10.5 \times 9.8 \times 2.4$ cm/$4.13 \times 3.86 \times 0.94''$<br>Power supply: $1.5$ V $\times 1$(AAA battery) |
| Moisture Content 0 to 50%<br>Accuracy $\pm$(5% + 5 digits) FS @23 $\pm$ 5 °C | Temperature measurement range: $-10$ °C ~ $+50$ °C<br>Temperature measurement accuracy: $\pm 1$ °C<br>Temperature resolution: 0.1 °C |
| Operating Temperature (0 to 50 °C)<br>Operating Humidity < 80% RH | Humidity measurement range: 10% RH–99% RH<br>Humidity measurement accuracy: $\pm 5$% RH<br>Humidity resolution: 1% |
| Max Resolution 0.1%<br>Dimensions $14.7 \times 1.6 \times 1.6''$ ($374 \times 40 \times 40$ mm)<br>Weight 9.4 oz (267 g) | |

The Independent Samples t-test was used to test the research question. The research question was: is there a difference in WCR, IT, T, and RH measurements between an Arduino microcontroller system and traditional devices? Therefore, the Hypotheses was:

**The null hypothesis (H0):** *There is no difference in mean WCR, IT, T, and RH measurements between an Arduino microcontroller system and traditional devices.*

**The alternative hypothesis (H1):** *There is a difference in mean WCR, IT, T, and RH measurements between an Arduino microcontroller system and traditional devices.*

The data analysis and building multiple mathematical models to predict the WRC were performed using the Design-Expert Version 13 software. The software was from Stat-Ease Inc., Minneapolis, MN, USA. The model followed the steps to building multiple regression models described in [26].

*2.5. Simulation Models*

A simulation model was necessary to measure the power and efficiency of any developed forecasting algorithm. It allowed the researchers to verify the robustness of the different algorithms by considering the component combination of the model to obtain the best test scenario. The model's performance was evaluated by the correctness of its estimation, its ability to reproduce the actual return in the simulation, and its stability. We wanted to regularly forecast the weather 14 weeks in advance and produce a new forecast each week. Hence, the estimated model in this study was validated and evaluated based on its forecasting power by using the mean square error (MSE), mean absolute percentage error (MAPE), and average accuracy percentage (AAP). The following performance measure functions were employed [27,28]:

1-The form of MSE can be written as follows:

$$\text{MSE} = \frac{1}{N} \sum_1^N (\text{Actual yield} - \text{Forecasted yield})^2$$

2-Mean absolute percentage error MAPE

$$\text{MAPE} = \frac{1}{N} \sum_1^N \frac{/(\text{Actual yield} - \text{Forecasted yield})/}{\text{Actual yield}} \times 100$$

3-Average accuracy percentage AAP

$$\text{AAP\%} = 100\% - \text{MAPE}$$

where: $N$ is the number of data points for $I = 1, 2, \ldots, N$.

## 3. Results and Discussion

*3.1. Comparison of the Arduino Microcontroller System and Traditional Devices*

WCR, IT, T, and RH levels were measured during both sunny and cloudy days by the Arduino microcontroller system and traditional devices in the greenhouse. The results were compared using an Independent Samples *t*-test as described in Table 3. Levene's test checked the null hypothesis that the variances of the two groups were equal. In this study, the p-value for WCR, IT, T and RH levels was 0.778, 0.589, 0.651, and 0.985, respectively. The assumption of equal variances was not violated so we can look at the top row of Table 3.

The values of the t statistic are 1.669, 1.251, 1.298, and 1.355, and the *p*-value is displayed as 0.107, 0.222, 0.206, and 0.187. This means that there is a very small probability of these results occurring by chance under the alternative hypothesis of difference between the two groups. The alternative hypothesis is formally rejected when accepting the null hypothesis. There is no difference in mean measurements between an Arduino microcontroller system and traditional devices. This means that there is very strong evidence that the WCR, IT, T, and RH levels for these two groups are no different, which can be seen clearly in Figures 4–7.

The Arduino microcontroller system was tested and compared with traditional measuring methods commonly used in greenhouses. Observed values of daily WCR, IT, T, and RH were obtained by means of WCR, IT, T, and RH sensors and readings on an LCD panel. The values could be downloaded onto a USB flash drive.

The daily WCR, IT, T, and RH were recorded for 100 days (14 weeks) and the weekly averages of WCR, IT, T, and RH were calculated. Figures 4–7 show the comparisons between weekly measurements of the WCR, IT, T, and RH from the Arduino microcontroller system

versus the traditional devices for 14 weeks during the planting season. The developed system was able to identify the behaviour and changes in the measured levels without significant differences as compared to the traditional methods. All these results indicate that the developed system was able to identify the environmental conditions and the amount and time that irrigation was used inside the greenhouses; these findings are in accordance with [17,19,20]. Based on the results obtained from the study (Table 3 and Figures 4–7), the Arduino microcontroller system has proven to be reliable when installed in large greenhouses and the system saves approximately 12.5% of the water normally used in these greenhouses. These observations agree with the results obtained by [11].

**Table 3.** T-test results of the independent samples for WCR, IT, T, and RH.

| | | Independent Samples Test | | | | | |
|---|---|---|---|---|---|---|---|
| | | Levene's Test for Equality of Variances | | *t*-Test for Equality of Means | | | |
| | | F | Sig. | t | df | Sig. (2-Tailed) | Std. Error Difference |
| WCR | Equal variances assumed | 0.081 | 0.778 | 1.669 | 26 | 0.107 | 0.15282 |
| | Equal variances not assumed | | | 1.669 | 25.823 | 0.107 | 0.15282 |
| IT | Equal variances assumed | 0.299 | 0.589 | 1.251 | 26 | 0.222 | 2.57028 |
| | Equal variances not assumed | | | 1.251 | 25.679 | 0.222 | 2.57028 |
| T | Equal variances assumed | 0.210 | 0.651 | 1.298 | 26 | 0.206 | 0.81842 |
| | Equal variances not assumed | | | 1.298 | 25.463 | 0.206 | 0.81842 |
| RH | Equal variances assumed | 0.000 | 0.985 | 1.355 | 26 | 0.187 | 1.87472 |
| | Equal variances not assumed | | | 1.355 | 25.997 | 0.187 | 1.87472 |

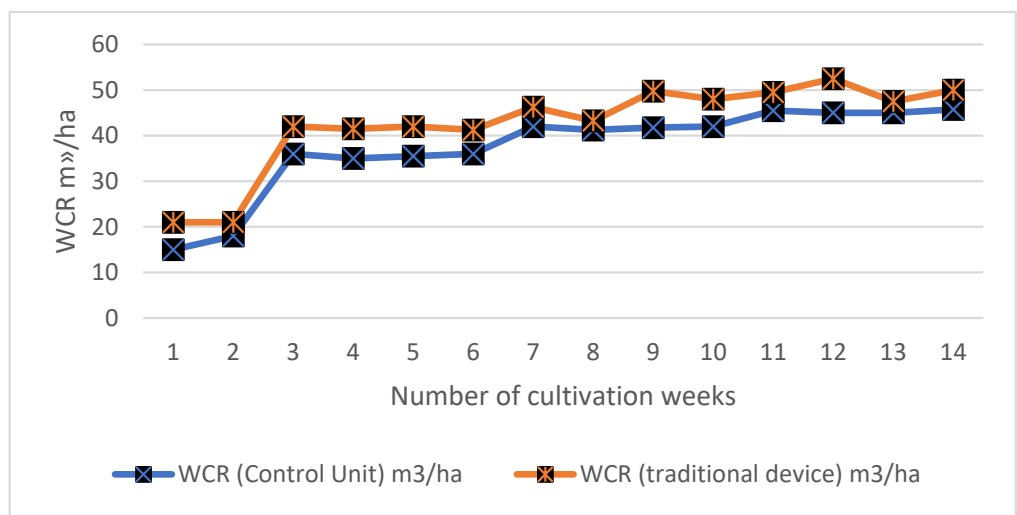

**Figure 4.** Results of WCR from an Arduino microcontroller system and traditional devices by week.

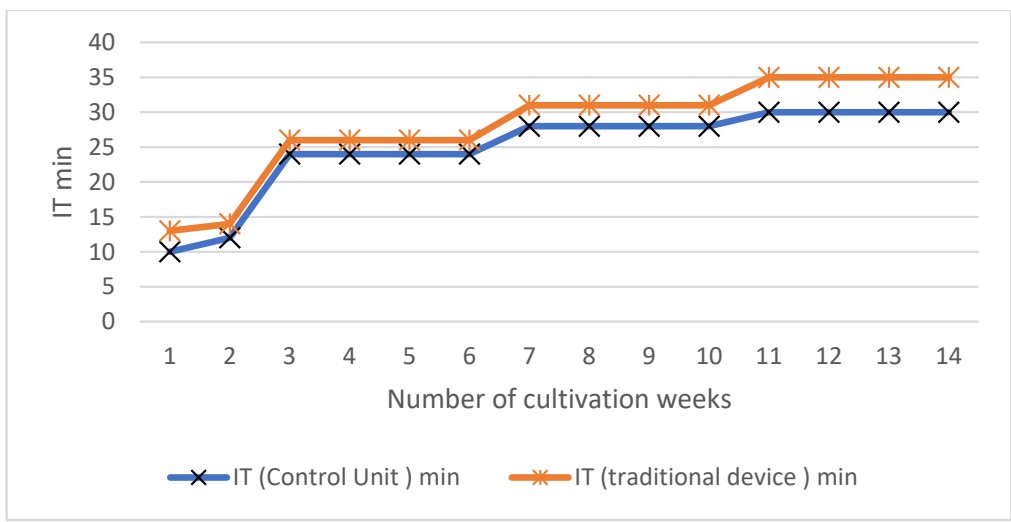

**Figure 5.** Results of IT from an Arduino microcontroller system and traditional devices by week.

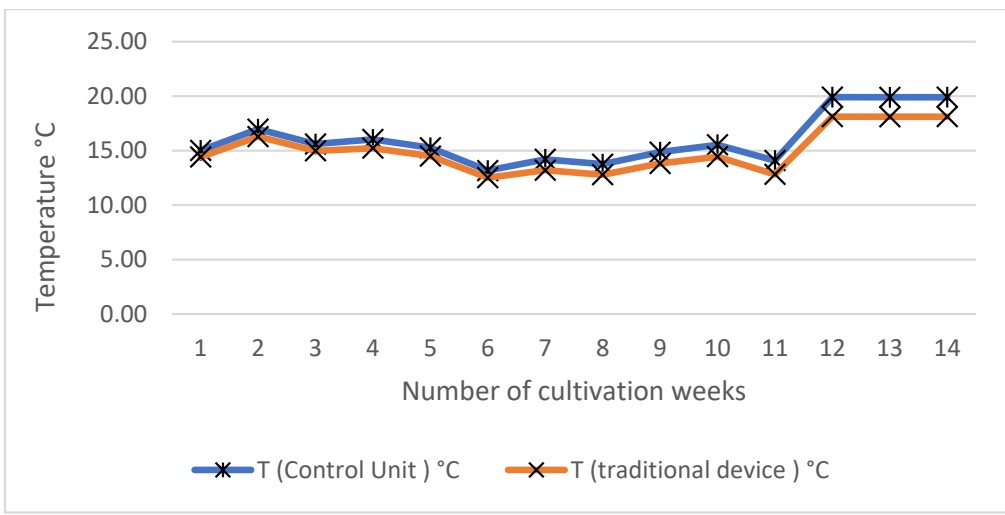

**Figure 6.** Results of T from an Arduino microcontroller system and traditional devices by week.

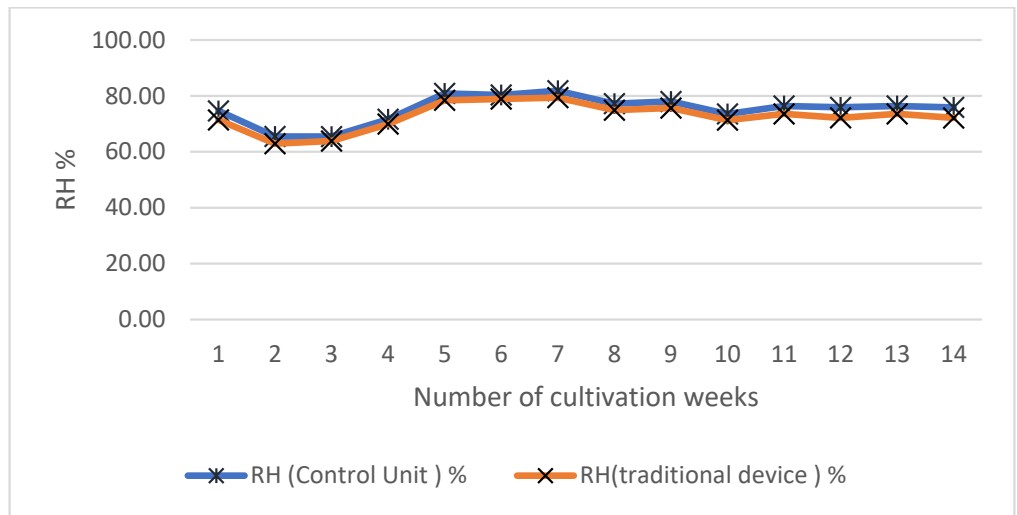

**Figure 7.** Results of RH from an Arduino microcontroller system and traditional devices by week.

### 3.2. Modelling and Relationships between WCR, IT, T, and RH

In Iraqi greenhouses, measurements of temperature, humidity, and irrigation duration that are collected by most commercial plantations are minimal since normally, only data concerning monthly whole rainy days and daily rainfall days are available [13]. Therefore, it is critical to provide a simplified methodology for weekly WCR using the limited temperature, humidity, and irrigation time data. The mathematical models used the T, RH, and IT data from the Arduino microcontroller system.

The resulting linear, 2FI, and quadratic equations and the corresponding values of R2 for the testing period of 14 weeks are presented in Table 4. The model equation with WCR, IT, T, and RH in weeks as the independent variables showed that R2 = 0.962, 0.969, and 0.977, implying that the model explained well 96.2, 96.9, and 97.7% of the variability in WCR. A linear relationship between actual and predicted weekly WCR showed a strong positive association with high significance at $p < 0.0001$.

**Table 4.** Mathematical models under an Arduino microcontroller system in a greenhouse.

| Model | F-Value | *p*-Value | $R^2$ | Adj. $R^2$ | Pred. $R^2$ | Std. Dev. |
|---|---|---|---|---|---|---|
| WCR = −0.0173451.0.001595T + 0.004261RH + 0.05329IT | 320.91 | <0.0001 | 0.962 | 0.959 | 0.952 | 0.093 |
| WCR = 8.12464 − 0.489734T − 0.119744RH + 0.052929IT + 0.007382T × RH − 0.002221T × IT + 0.000465RH × IT | 184.02 | <0.0001 | 0.969 | 0.964 | 0.954 | 0.087 |
| WCR = 20.54747 − 1.17731T − 0.275757RH − 0.012634IT + 0.012368T × RH − 0.000944T × IT + 0.001470RH × IT + 0.008004T$^2$ + 0.000328RH$^2$ − 0.000588IT$^2$ | 155.26 | <0.0001 | 0.977 | 0.971 | 0.959 | 0.077 |

The F-test of overall significance indicates whether a model provides a better fit to the data than a model that contains no independent variables. Model F-values of 320.91, 184.02, and 155.26 and *p*-values less than 0.0500 indicate that the model terms are significant. The F-values with a probability of 0.0001 indicate that the regression coefficients are nonzero. There is only a 0.01% chance that F-values these large could occur due to noise. The predicted $R^2$ of 0.9528, 0.9542, and 0.9598 are in reasonable agreement with the Adjusted $R^2$ of 0.9590, 0.9640, and 0.9713, i.e., the difference is less than 0.2 for linear, 2FI, and quadratic models, respectively.

Figure 8 was a scatter plot that showed the correlation between predicted values and actual values for linear, 2FI, and quadratic models to see how the models performed with the mean and best values of the variables in every prediction. Figures graphically represent an association of the actual and predicted WCR for all models. The magnitude of error expressed as the difference between predicted $R^2$ and adjusted $R^2$ values is also shown in the same Table 4.

The practical predictive ability of the linear, 2FI, and quadratic models was visual, and the representation strongly suggests a goodness of fit for models in predicting WCR; the magnitude of error was very small. The mean error is preferred for the whole population rather than a single sample [29]. According to Figure 8A, the linear predicted model fully fit (without deviations) to the actual values and the $R^2$ value is 0.962. Reference [26] explained that the best model has an R-square value above 80% and it uses the smallest number of parameters.

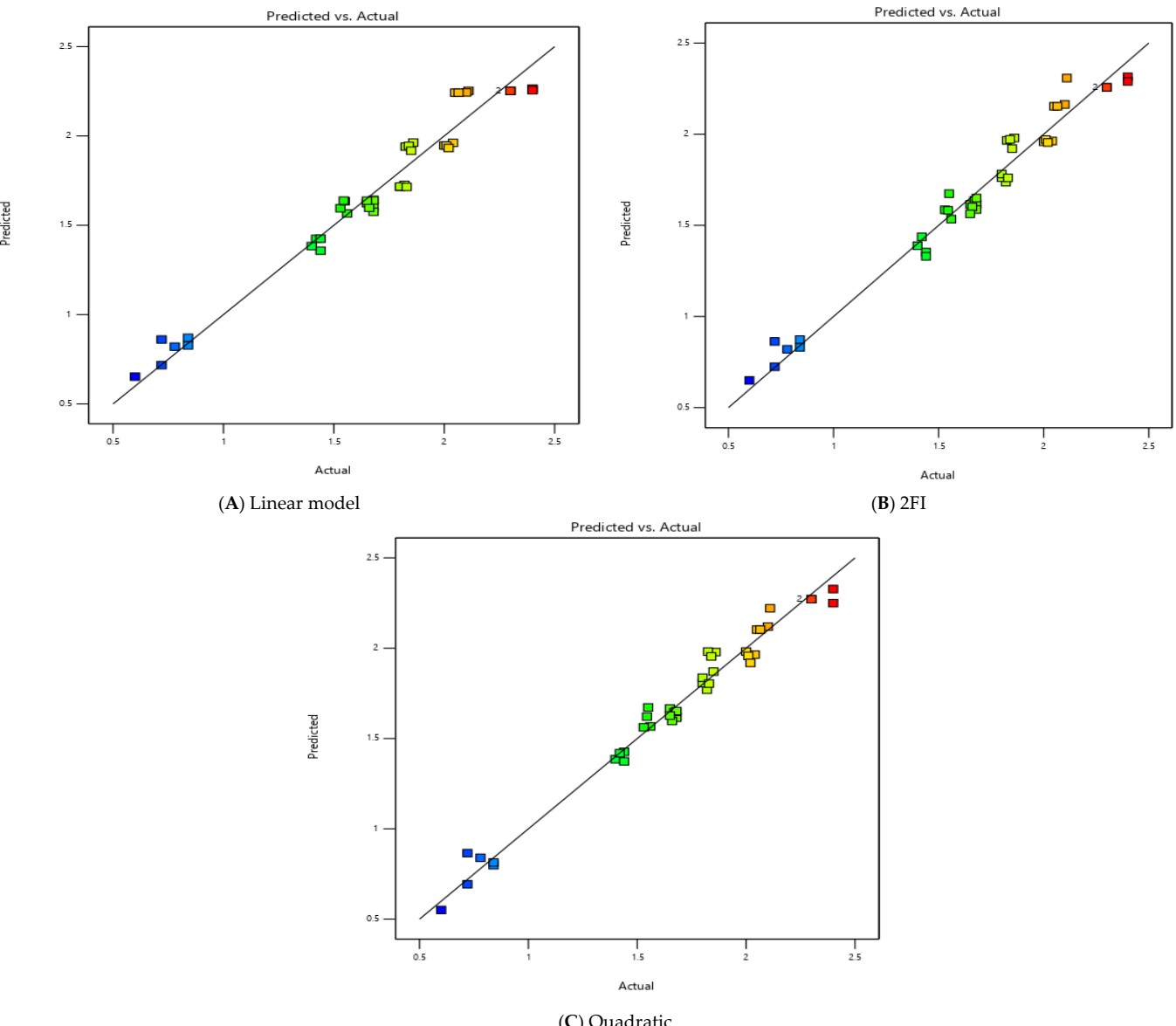

(**A**) Linear model

(**B**) 2FI

(**C**) Quadratic

**Figure 8.** The scatter plot of the predicted vs. actual values for linear, 2FI, and quadratic models. (**A**) Linear model (**B**) 2FI model (**C**) Quadratic model.

The effect of WCR varies with the degree of temperature, humidity, and irrigation duration for the cucumber plants. Water deficit stress during pre-flowering and grain filling stages massively affects plant performance due to the imprecise traits function. Water stress increased the flowering days and days to maturity while it decreased the leaf number and led to a loss of normal root architecture which further led to a reduction in yield [30]. The changes in WCR due to temperature, humidity, and irrigation duration variations usually contribute to increasing yield [17,31]. Overall, our present study indicated that WCR has a strong relationship with temperature, humidity, and irrigation time. These factors accounted for increasing the $R^2$ of the models as WCR is dependent on temperature, humidity, and irrigation duration.

Table 5 provides a comparison between the linear forecasts and the actual values based on the MSE, MAPE, and AAP within the studied states. The results were obtained for the forecasted WCR for 14 weeks. The outputs of the linear model were found to be closer to the actual values for the WCR. The finding indicated that the average accuracy percentage for the forecasts by the linear models had a record-high value of 92.038. In contrast, the importance of the MSE and MAPE for the model was lower. Therefore, the linear model

was appropriate for forecasting the data and can be used as an alternative model to predict the WCR.

**Table 5.** Simulation models of WCR in greenhouses.

| Weeks | Actual WCR $m^3$ | Predicted WCR $m^3$ |
|---|---|---|
| 1 | 0.6 | 0.589 |
| 2 | 0.72 | 0.623 |
| 3 | 1.44 | 1.285 |
| 4 | 1.4 | 1.304 |
| 5 | 1.42 | 1.357 |
| 6 | 1.44 | 1.391 |
| 7 | 1.68 | 1.592 |
| 8 | 1.65 | 1.580 |
| 9 | 1.67 | 1.564 |
| 10 | 1.68 | 1.534 |
| 11 | 1.82 | 1.677 |
| 12 | 1.8 | 1.574 |
| 13 | 1.8 | 1.576 |
| 14 | 1.83 | 1.576 |
| MSE % | 0.019 | |
| MAPE % | 7.961 | |
| AAP % | 92.038 | |

## 4. Conclusions

The main objective of an optimal environment and evaluation is to control critical parameters such as water consumption, irrigation time, temperature, and humidity, based on the design of a microcontroller system in a greenhouse which is a purely sensor-based system. The results show that the developed system maintained WCR, IT, T, and RH levels and reduced water consumption by approximately 12.5%. We hope that the new system will be used as a prototype for Iraqi greenhouses to automatically control and monitor WCR, IT, T, and RH levels.

The developed models are simple, efficient, and easily applied. The models indicated that the regression equation well represented 96.9% of the variability in weekly WCR. A linear relationship between actual and predicted weekly WCR shows that the correlation = 0.984, which is a strong positive association. Additionally, the linear model was chosen as the best model to use in greenhouses, with the average accuracy percentage simulation and mean square error being 92.038 and 0.01988%, respectively. The researchers suggest conducting experiments with other crops under the same conditions to verify the accuracy of the proposed equations. Future research could develop a raspberry pi microcontroller system and compare it with the Arduino microcontroller system. Additionally, the Arduino microcontroller system used in this study could be applied to other types of crops, thus increasing the general applicability of the method presented in this article.

**Author Contributions:** Conceptualization, Y.Y.H., M.K.K., J.v.D. and K.M.; Methodology, Y.Y.H. and M.K.K.; Software, Y.Y.H.; Validation, Y.Y.H., M.K.K., J.v.D. and K.M.; Formal analysis, Y.Y.H.; Investigation, Y.Y.H.; Resources and data curation, Y.Y.H. and M.K.K.; Writing—original draft preparation, Y.Y.H.; Writing—review and editing, J.v.D.; Visualization, supervision; project administration and funding acquisition, K.M. All authors have read and agreed to the published version of the manuscript.

**Funding:** This study was funded by Nuffic, the Orange Knowledge Programme, through the OKP-IRA-104278 project titled "Efficient water management in Iraq switching to climate smart agriculture: capacity building and knowledge development" coordinated by Wageningen University & Research, The Netherlands.

**Institutional Review Board Statement:** Not applicable.

**Informed Consent Statement:** Not applicable.

**Data Availability Statement:** Not applicable.

**Acknowledgments:** The authors wish to express their appreciation to Mosul University, Nuffic and Wageningen University for funding this research under the OKP-IRA-104278 project.

**Conflicts of Interest:** The authors declare no conflict of interest. The funders had no role in the design of the study; in the collection, analyses, or interpretation of data; in the writing of the manuscript, or in the decision to publish the results.

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
