# Peer review of "Automatic Water Control System and Environment Sensors in a Greenhouse"

_water, doi:10.3390/w14071166_

Round 1

Reviewer 1 Report

Dear author’s

I was pleased to review your manuscript “Automatic Water Control System and Environment Sensors in a Greenhouse“ The topic is interesting.

However, the manuscript should be further improved. In order to improve the manuscript a have the following comments.

Abstract. I recommend you to revise your abstract:

There are multiple information in the abstract and repeated in the introduction section.

The aim: I recommend you to specify the aim of your experiments.

Introduction section should introduce the reader with general concept of the manuscript. There are multiple information about agriculture. Please keep only important info related to water control system and environment sensors in a greenhouse.

Please explain the novelty of your study.

Discussion - in this section is mandatory to compare your experiments with the literature. Please specify the limitations of your study.

Conclusion: Please do not repeat the same info belong the section of the article.

I recommend you to remove “The development of a microcontroller system for controlled climatic conditions in a green-395 house environment with efficient production forms an important part of the agriculture sector.”

Please modify the references accordingly.

Punctuations and English reviews are necessary;

Reviewer 2 Report

Dear Authors,

Following are some of the comments:

1) Currently, the title paper title needs to be changed as it is not in line with the contents of the paper.

2) The Introduction section needs to answer following questions what is the issue, why is it an issue, how in the past/or currently this issue is being addressed, and how you intend to address this issue. Finally, add the structure of the paper. Add following papers: a) An Internet of Things Approach for Water Efficiency: A Case Study of the Beverage Factory b) Improving Water Efficiency in the Beverage Industry with the Internet of Things 

3) Materials and Methods: This section needs restructuring and more focus on data collection and techniques for data collection.

4) Results and discussion section - Section 3.1 - Authors need to define which traditional devices  and traditional methods are they comparing it against.

5) Conclusion needs to highlight the limitations and recommendations for future research.

6) Author contributions - Use only author initials 

7) Reference list - Mention the doi's for each of the article.

Round 2

Reviewer 1 Report

Dear author's

Thank you for your revise version of your article. Your experiment is interesting and i think that your article can be published.

Best regards, 

Reviewer 2 Report

The authors have addressed most of the corrections and have considerably improved the manuscript.